Journal: https://www.earth-system-science-data.net/

**Water masses distribution offshore the Sabrina Coast (East Antarctica)**
Bensi Manuel[1], Kovačević Vedrana[1], Donda Federica[1], O'Brien Philip Edward[2], Armbrecht Linda[3], Armand Kay Leanne[4]
[1] National Institute of Oceanography and Applied Geophysics (OGS), Trieste, 34010, Italy;
[2] Department of Environmental Sciences, Macquarie University, Sydney, Australia;
[3] Institute for Marine and Antarctic Studies, University of Tasmania, Battery Point TAS 7004, Australia;
[4] Australian National University, Canberra, Australia;
*Correspondence to*: M. Bensi (mbensi@inogs.it)
**Abstract**
Current glacier melt rates in West Antarctica substantially exceed those around the East Antarctic margin. The exception is Wilkes
Land where, e.g., Totten Glacier, underwent significant retreat between 2000 and 2012, underlining its sensitivity to climate
change. This process is strongly influenced by ocean dynamics, which in turn changes in accordance with the evolution of the ice
caps. Here, we present new oceanographic data (temperature, salinity, and dissolved oxygen) collected during austral summer
2017 offshore the Sabrina Coast (East Antarctica) from the continental shelf break to ca 3000 m depth. This area is characterized
by very few oceanographic in situ observations.
The main water masses of the study area, identified by analysing thermohaline properties, are: the Antarctic Surface Water with
potential temperature $\theta > -1.5$ °C and salinity $S < 34.2$ ($\sigma_\theta < 27.55$ kg m$^{-3}$), the Winter Water with $-1.92 < \theta < -1.75$ °C and $34.0$
$< S < 34.5$ (potential density, $27.55 < \sigma_\theta < 27.7$ kg m$^{-3}$), the modified Circumpolar Deep Water with $\theta > 0$ °C and $S > 34.5$ ($\sigma_\theta >$
$27.7$ kg m$^{-3}$), and Antarctic Bottom Water with $-0.50 < \theta < 0$ °C and $34.63 < S < 34.67$ ($27.83 < \sigma_\theta < 27.85$; neutral density $\gamma^n >$
$28.30$ kg m$^{-3}$). The latter is a mixture of dense waters from the Ross Sea and Adélie Land continental shelves. Such waters are
influenced by the mixing processes they undergo as they move westward along the Antarctic margin, also interacting with the
warmer Circumpolar Deep Water.
The spatial distribution of water masses offshore the Sabrina Coast also appears to be strongly linked with the complex morpho-
bathymetry of the slope and rise area, supporting the hypothesis that downslope processes contribute to shaping the architecture
of the distal portion of the continental margin.

**Short summary (plain text)**
The Totten Glacier (Sabrina Coast, East Antarctica) has undergone significant retreat in recent years, underlining its sensitivity to
climate change and its potential contribution to global sea-level rise. The melting process is strongly influenced by ocean dynamics,
while the spatial distribution of water masses in this region appears to be linked to the complex morpho-bathymetry, which also
supports the hypothesis that downslope gravity currents contribute to shaping the architecture of the continental margin.

## 1 Introduction

[1] Polar regions are key components of Earth's climate system and are particularly sensitive to ongoing climate change effects
induced by anthropogenic pressures. It has been estimated that the full melting of all Antarctic ice has a sea level equivalent (SLE)
of ~58 m (Fretwell et al., 2013).
Understanding Earth's climate processes as well as their future projections depends on the constant collection and interpretation
of long-term scientific data and palaeoclimatic records. Therefore, studying long-term records from polar regions is key (e.g.,
Masson-Delmotte et al., 2013) to have a more complete understanding of the region's past climate variability (e.g., Escutia et al.,
2019).  Within this frame, quantifying sea-level rise associated with global warming is crucial, and the accuracy of such
quantifications ultimately depends on our knowledge of the response of polar regions to global warming. In fact, there is still
significant uncertainty in estimates around sea-level rise, as Church et al. (2013) reported in the IPCC AR5 Sea Level Chapter.
They delineate that significant challenges remain in understanding and predicting processes related to the dynamic response of
Antarctic marine-terminal glaciers and marine sectors. The Antarctic ice sheet response to current climate forcing can be elucidated
by examining how the ice sheet had behaved in response to similar climate forcings in the past. In spite of the significant effort in
putting the estimates together, the IPCC report expresses only medium confidence in estimating the contribution of Antarctic ice
melt to sea levels during the last major warm episode, e.g. the Last Interglacial period, ca 129k to 116k years ago (Masson-
Delmotte et al., 2013).
Marine processes change in accordance with the evolution of the ice caps and vice versa, especially in those areas where glaciers
are grounded below sea level. This condition occurs not only in West Antarctica, but also in several portions of East Antarctica
(e.g., Adelie Land, Rignot et al., 2011), including the Sabrina Coast. Here, the Totten Glacier, the downstream end of the vast
Aurora subglacial basin, is largely grounded below sea level and hence susceptible to rapid ocean-driven ice sheet basal melting
(Pritchard et al., 2009; Roberts et al., 2011; Young et al., 2011; Rignot et al., 2013; Aitken et al., 2016; Hirano et al., 2021). Totten
Glacier is, indeed, exposed to temperatures up to 3 °C above the ice shelves' melting point (Rintoul et al., 2016). The Aurora
subglacial basin today also hosts an active subglacial hydrological system that drains basal meltwater to the ocean (Wright et al.,

55  2012).

[2] According to Silvano et al. (2018), while relatively warm waters (> 0°C) flood the continental shelves in West Antarctica
driving rapid basal melt of ice shelves, the ice shelves in East Antarctica experience relatively low rates of basal melt because they
are influenced by cooler waters (< 0°C). However, the Totten Glacier is an exception, since the melting rates of this glacier and of
the nearby Moscow University ice shelf (MUIS) are among the fastest in the East Antarctic Ice Sheets (Khazendar et al., 2013; Li
et al., 2015; Mohajerani et al., 2018): the glacier draining into MUIS shows a 3 Gt/y loss in 1979–2003 and a 0.3 Gt/y gain in

2017, whereas Totten Glacier loss has increased through time from 5.7 Gt/y in 1979–2003 to 7.3 Gt/y in 2003–2017 (Rignot et al., 2019). These changes are enhanced by incursions of relatively warm modified Circumpolar Deep Water (mCDW) to the continental shelf and to the glacier grounding line, favoured by wind stress, local eddies, and bathymetric constraints (Rintoul et al., 2016; Silvano et al., 2016; 2017; Nitsche et al., 2017; Greene et al., 2017; Hirano et al., 2021). In fact, a deep and extensive pool of water with maximum temperature of ~0.7°C has been identified at the outer continental shelf over a wide bathymetric depression at depths below 400–500 m (Nitsche et al., 2017). Ice loss can locally be also favoured by low levels of sea ice and Dense Shelf Water production in the Antarctic polynyas (Tamura et al., 2008), although there is still no clear evidence, surprisingly, of dense shelf water production in the Dalton polynya, east of the Totten ice shelf (Silvano et al., 2018). Rignot et al. (2019) reported that Totten Glacier holds an ice volume that translates into a SLE of 3.85 m. The mCDW, being the major heat source on the Sabrina Coast continental shelf, is characterized by a temperature larger than −0.4 °C and a salinity around 34.5–34.6 (Silvano et al., 2017), and comprises the bottom layer of the water column. This appears to be different from other coastal areas of East Antarctica, where the bottom layer is usually occupied by colder and denser Shelf Water (Bindoff et al., 2000; Greenbaum et al., 2015). Studies of rapidly retreating outlet glaciers in the Amundsen Sea have shown a major role of the mCDW in transporting heat from the deep ocean onto the shelf, leading to enhanced glacier melting (Smith et al., 2011; Pritchard et al., 2012). This process has also been suggested as a possible cause for the rapid melting of Totten Glacier (e.g., Williams et al., 2011; Pritchard et al., 2012; Greene et al., 2017), but other mechanisms have been also suggested. Pritchard et al. (2012) invoked that changes in the circum-Antarctic wind field and its effects on upwelling around the continent cause the rapid basal melting of the Totten Glacier. Also, Khazendar et al. (2013) and Gwyther et al. (2014) have argued that this enhanced melting is the result of complex interactions between oceanic and shelf water masses at the base of the ice. In particular, Gwyther et al. (2014) suggested that the ice melting varies on seasonal and interannual timescales, with increased ice melt of the Totten Glacier coinciding with a reduced strength of the nearby Dalton polynya. These studies rely heavily on satellite observations and bathymetry of the regions, e.g., the 1 km horizontal grid resolution obtained through the General Bathymetric Chart of the Oceans (GEBCO) used for oceanographic models (Gwyther et al., 2014). The GEBCO grid has 2 ship tracks in the critical 150 km by 200 km area seaward of the Totten Glacier, meaning that most of the grid is a "best guess".

[3] The Antarctic ice sheet nucleated in the higher elevations of the Gamburtsev Mountains and first reached the ocean near the Sabrina Coast and Prydz Bay (Huybrechts 1993; DeConto & Pollard 2003). Marine-terminating glaciers existed at the Sabrina Coast by the early-middle Eocene (ca. 56–41 Ma), implying the occurrence of ice caps before the emplacement of continental-scale ice sheets (Gulick et al., 2017). The first preserved evidence of grounded ice on the Sabrina Coast shelf is suggested to be Late Eocene in age (ca. 38 Ma; Gulick et al., 2017). The climate evolution of the Wilkes Land margin, and thus of the Sabrina Coast, from the pre-glacial Era to the present time comprises three main key periods: i) pre-glacial conditions (Phase 1 reported in Donda et al., 2007); ii) growth and development of a polythermal, highly dynamic ice sheet (Phases 2 and 3 in Donda et al., 2007), and iii) transition to polar conditions (Phase 4 in Donda et al., 2007). This overall picture is well supported by several Antarctic and sub-Antarctic stratigraphic records, e.g., the Deep Sea Drilling Program Leg 28, Cape Roberts Drilling Project, Ocean Drilling

Program Legs 119, 188, and 189 and International Ocean Discovery Program Leg 318 (Donda et al., 2020). Compared to other
areas, the Sabrina Coast seismostratigraphy suggests that downslope processes contributed to shaping the distal margin architecture
even during its latest development stages (i.e., Phase 4), when glacial meltwater-related fluxes were able to erode and deliver
sediments to the rise area (Donda et al., 2020). The continental slope and rise of the Sabrina Coast reveal remarkable differences
between the eastern and western areas, as also highlighted by the present-day bathymetry (Fig. 1). The western sector is shallower
and characterized by the presence of two prominent NE-SW trending ridges, separated by a low sinuosity narrow submarine
canyon flanked with terraces. Here, one of these bathymetric highs favours the generation of  a cyclonic gyre centered on 115°E,
just north of the shelf break (Wakatsuchi et al., 1994). Armand et al. (2018) described the eastern facing slopes of the ridges (Fig.
1) as smooth seafloor with significant evidence of mass movement (e.g., slump scars and debris runout fields), while their western
facing slopes are gullied terrain (with a gully depth range of up to 15 m). In fact, the associated canyon is not receiving a high
proportion of downslope turbidity currents, rather being fed by slumping on the adjacent ridge flanks (O'Brien et al., 2020). The
eastern ridge was reported as having a lower slope than the western ridge, and being dominated by slump scars.
The eastern sector, instead, is characterized by a complex network of erosional channels and it is shaped by dendritic canyons,
which meander and bend sharply to then join less sinuous main channels, the floors of which contain terraces and closed
depressions (O'Brien et al., 2020). The ridges between canyons are clearly tied to their adjoining canyons and formed by westward
advection of fine sediment lofted from turbidity currents and deposition of pelagic sediment (O'Brien et al., 2020). The western
and the eastern areas are separated by a broad depression linked with one of the dendritic canyons near the lower slope. The upper
slope consists of a smooth to gullied apron, downslope of which the canyon heads begin. The shelf break occurs at depths of 480
- 510 m.
[4] This paper aims at describing the thermohaline structures from the continental shelf break to about 3000 m depth identified in
the offshore area along the Sabrina Coast, by analysing physical oceanographic data collected in 2017 during a multidisciplinary
Antarctic expedition (see below).

**2 The oceanographic dataset: instrumentation and data processing**

Oceanographic data presented in this paper were collected offshore the Sabrina coast, between 113°E and 122°E and 66° S and 64°S (Fig. 1) during the marine geoscience expedition "Interactions of the Totten Glacier with the Southern Ocean through multiple glacial cycles" (IN2017_V01) (see Armand et al., 2018). This expedition took place between 14 January and 7 March 2017 on board the Australian Marine National Facility (MNF) Research Vessel *Investigator.*

.

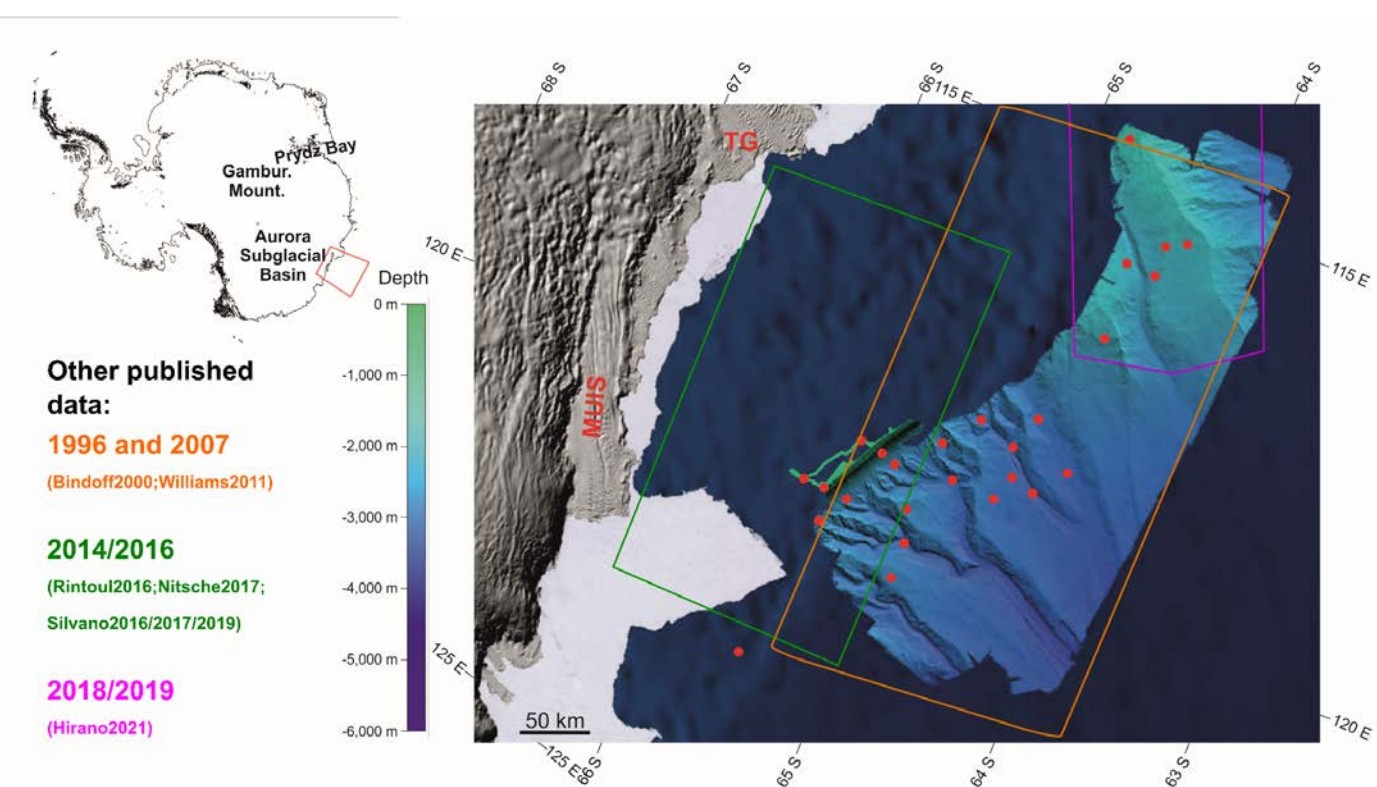

*Figure 1 - Sabrina Seafloor Survey area. Red dots indicate CTD casts collected during the IN2017_V01 cruise. Coloured polygons, instead, indicate areas where other already published hydrological data have been taken (reference years and related articles are indicated on the left). The high resolution bathymetry obtained from multibeam data acquired during the IN2017_V01 cruise (from O'Brien et al., 2020) is superimposed over the lower resolution bathymetry from the International Bathymetric Chart of the Southern Ocean. On land, the terrain map comes from the Reference Elevation Model of Antarctica (REMA, Howat et al., 2019). The colorbar on the left refers to the high resolution multibeam bathymetry.*

We take into account 31 Conductivity-Temperature-Depth (CTD) vertical profiles (labeled 1-11, 13-23, 25-33) acquired in the study region (Fig. 2) using a Seabird SBE 9plus (pump-controlled) CTD and SBE 11plus V2 Deck Unit. The CTD was integrated with a SBE32 Carousel Water Sampler equipped with 36 Niskin bottles (OceanTest Equipment Inc. Florida; 12-L capacity each one, mounted on the rosette sampler), with additional sensor for measuring dissolved oxygen concentration (DO, SBE43), and

altimeter (PA500). The Commonwealth Scientific and Industrial Research Organisation (CSIRO) supplied calibration factors that were used to compute pressure, temperature, and conductivity/salinity values. Data were subjected to automated quality control (QC) to remove spikes and out-of-range values (see https://www.marine.csiro.au/data/reporting/get_file.cfm?eov_pub_id=1512, last access on October 26, 2021), while maintaining true data features. An additional filter was applied to the data to evaluate the median and standard deviation of the conductivity over a moving window. This has made it possible to detect extreme changes in the sensor values characteristic of the noise induced by spikes. The conductivity calibration was based on two deployment groupings, due to sensor changes during the voyage, and it was based upon the comparison between conductivity data obtained from CTD and water samples (104 of the total of 151 water samples taken during deployments). The final calibration for casts 1-13 from the secondary sensor had a standard deviation (std) of 0.001 psu, while the final calibration for casts 14-33 from the secondary sensor had a std of 0.002 psu. Water samples were also collected and used to compute new estimates of DO calibration coefficients, obtained by applying a linear regression. A single calibration group from each sensor was used with the associated SBE43 up-cast data. The DO calibration had a std of 0.85911 $\mu$M with a good agreement between the sensor and bottle data. Based on the results obtained from the calibration procedure, the final dataset was obtained after 1 dbar binned averaged data from the secondary sensors (primary sensor for DO). Note that seven CTD profiles (casts 1, 2, 11, 13, 14, 15, and 18, see Fig. 2a) stopped before reaching the seafloor (between 200 and 700 m, e.g., for testing the new sensor settings). Potential temperature ($\theta$, °C) and potential density anomaly ($\sigma_\theta$, kg m$^{-3}$, referred to 0 dbar), and neutral density ($\gamma^n$, kg m$^{-3}$) were calculated using the toolbox TEOS-10 (http://www.teos-10.org/software.htm). We use $\sigma_\theta$ when considering the property distribution in the upper 500 m, and where the specific $\sigma_\theta$ values determine and delimit specific water masses within the upper layers. However, the depth range of our study area is wide, and $\sigma_\theta$ errors augment with depth, therefore $\gamma^n$ for the entire vertical range along vertical sections is used instead to depict the property distributions. Neutral density ($\gamma^n$) is a function of temperature, salinity, pressure, latitude and longitude, where the reference level is slightly adjusted at each point to compensate for the nonlinearity of the equation of state, hence it can be considered a locally-referenced potential density. Some figures were created using Ocean Data View (ODV; Schlitzer, 2021). More detailed information on instrumentation and quality control procedure are available at https://www.marine.csiro.au/data/trawler/survey_details.cfm?survey=IN2017_V01 (last access on October 26, 2021).

Finally, satellite images (MODIS - Moderate-resolution Imaging Spectroradiometer, Corrected Reflectance imagery) were used to highlight both the evolution of sea ice within the period covered by the IN2017_V01 cruise and the extension of the Dalton Polynya (Fig. 2b,c), the open water surrounded by sea ice in the vicinity of Totten Glacier and MUIS. This is one of the largest Antarctic coastal polynya, with its wintertime average area of $3.7 \pm 2.0$ $10^3$ km$^2$ ($6.5$ $10^3$ km$^2$ at the time of the cruise; Fig. 2b,c), extending in the prevalent downwind direction (see e.g., Arroyo et al., 2019). Satellite images used in this work are freely available from the NASA Worldview application (https://worldview.earthdata.nasa.gov), part of the NASA Earth Observing System Data. The spatial imagery resolution is 250 m, and the temporal resolution is daily.

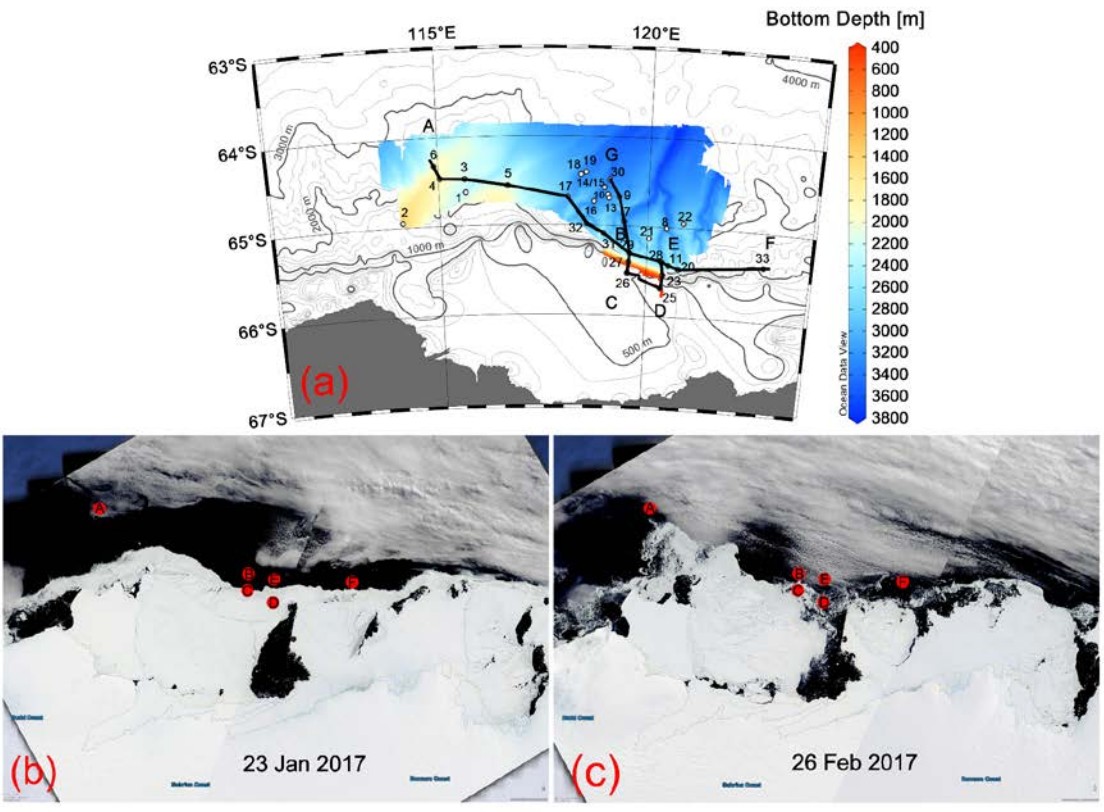


***Figure 2*** *- CTD stations map and color-coded high resolution bathymetry collected during the* IN2017_V01 *cruise (from O'Brien*
*et al., 2020), superimposed over the International Bathymetric Chart of the Southern Ocean (contour lines). Positions A, B, C, D,*
*E, F, and G denote along-slope and cross-slope transects (a). The same positions A-F (red dots) along the ice edge are*
*superimposed on satellite images (MODIS Corrected Reflectance imagery freely available from*
*https://worldview.earthdata.nasa.gov/) taken on 23 January (b) and 26 February 2017 (c), which show the temporal evolution of*
*the sea ice extension in the study region at the beginning and at the end of the IN2017_V01 cruise.*
**3 Thermohaline patterns in the study region**
**3.1 Typical water masses**
CTD casts are distributed over the continental slope and rise, offshore the area delimited by Totten Glacier and MUIS (Fig. 2).
The main water masses are identified by analysing their θ-S (and DO) properties (Fig. 3) and classified according to Silvano et al.
(2017, 2020). They are: Antarctic Surface Water (AASW) with θ > -1.5 °C and S < 34.2 ($\sigma_\theta$ < 27.55), Winter Water (WW) with -
$1.92 < \theta < -1.75$ and $34.0 < S < 34.5$ ($27.55 < \sigma_\theta < 27.7$ kg m$^{-3}$), and mCDW, with $\theta > 0$ °C and S > 34.5 ($\sigma_\theta > 27.7$ kg m$^{-3}$). The
AASW and WW are the most ventilated and therefore have the highest dissolved oxygen values. Acquired data do not reveal the
presence of water with characteristics either of Ice Shelf Water (ISW, $\theta < -1.92$ °C, $S < 34.5$, $27.55 < \sigma_\theta < 27.7$ kg m$^{-3}$) or Dense
Shelf Water (DSW, $\theta < -1.92$ °C, $S > 34.5$, $\sigma_\theta > 27.7$ kg m$^{-3}$). In particular, the former is found close to the two ice shelves, and
the latter forms within the polynyas, because of the intense cooling and brine rejection processes that can take place there. AASW
comprises a wide range of both temperature and salinity. It is warm due to summer heating, and fresh due mainly to sea ice melting.
The WW occupies a homogeneous layer of the water column formed as a result of convection on the shelf during the previous
winter. The mCDW is a relatively warm and salty water mass that can access the outer continental shelf through a section of the
shelf break over 500 m depth (Nitsche et al., 2017), reaching the ice shelf cavity through a recently discovered oceanic entry route
(Greenbaum et al., 2015) and causing ice melt. The thermohaline properties associated with the densest waters are attributed to
the Antarctic Bottom Water (AABW, with $-0.50 < \theta < 0$ °C, $34.63 < S < 34.67$, $27.83 < \sigma_\theta < 27.85$ kg m$^{-3}$, and $\gamma^n > 28.30$ kg m$^{-3}$).
The AABW in this part of the Australian Antarctic Basin (AAB) is a mixture of the local and remote dense waters, namely the
Adélie Land Bottom Water (ALBW) and Ross Sea Bottom Water (RSBW, see e.g., Silvano et al., 2020), both of which have
distinct characteristics in their source regions (Thomas et al., 2020). The most recent typical θ-S average values encountered during
2018 in the AAB  were -0.599 °C/34.704 for the RSBW, and  -0.632 °C/ 34.619 for the ALBW (Thomas et al., 2020). We argue
that these endmember values are representative also for the conditions encountered in 2017. However, thermohaline properties
reported for the AABW in 2017 in our study area (Fig. 3) were slightly higher (~ +0.10 °C and +0.05 for θ and S, respectively)
than those referred to the mentioned endmembers, reflecting the mixing process that bottom waters experience as they move
westwards along the Antarctic margin far from their area of origin.



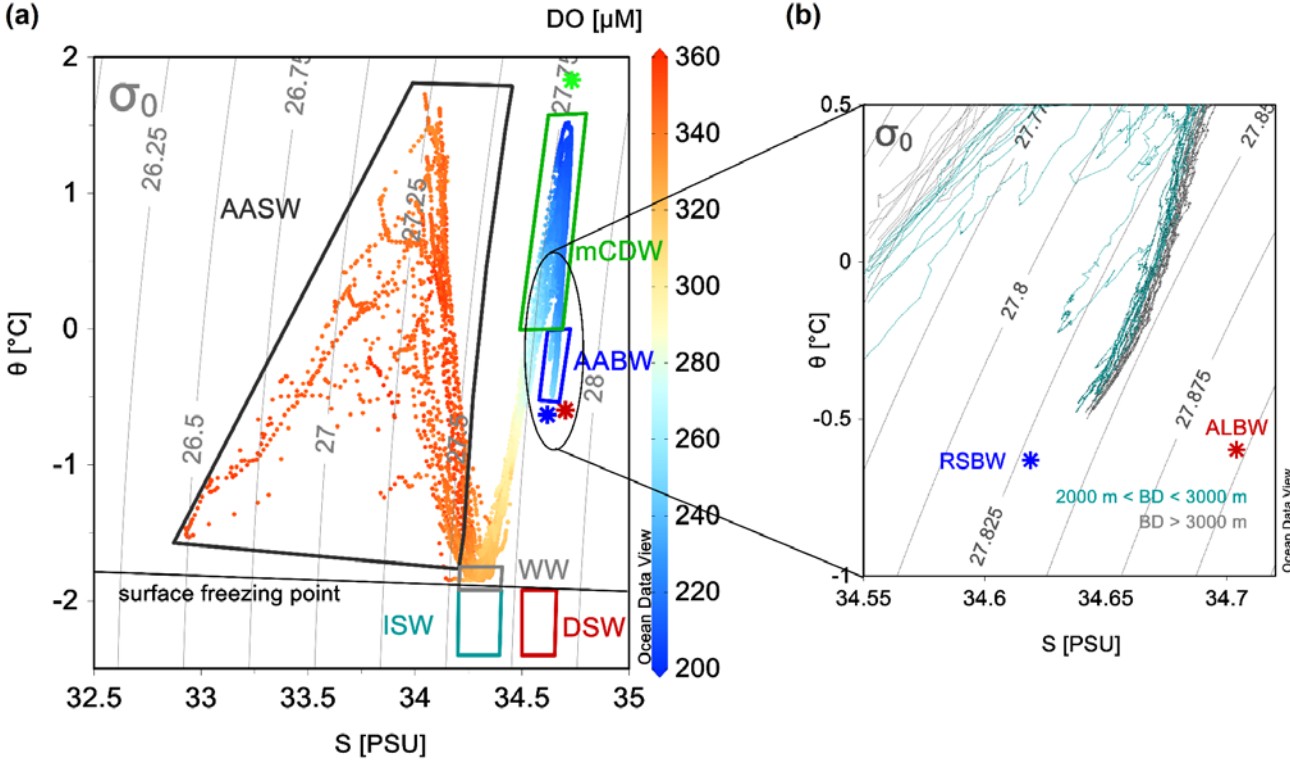


*Figure 3 - θ-S diagram (a) with colours referring to dissolved oxygen concentration (colorbar on the right). Characterization of the principal water masses, based on the continental shelf data according to Silvano et al. (2017, 2020): AASW - Antarctic surface water, WW - winter water, mCDW - modified Circumpolar Deep Water, AABW- Antarctic bottom water (with properties captured at 150°E), ISW - Ice Shelf Water and DSW - Dense Shelf Water. Zoom of the θ-S diagram (b) into the deepest layers where Bottom Depths (BD) are larger than 2000 m. Endmembers of RSBW - Ross Sea Bottom Water (red asterisk), ALBW - Adélie Land Bottom Water (blue), and CDW (green) are indicated (Thomas et al., 2020).*

## 3.2 Spatial distribution of the hydrographic properties

To describe thermohaline properties in the study area and in proximity of the sea ice edge, we consider a zonal section (West-Est) running almost parallel to the continental slope, extending from station 6 to station 25 (Fig. 4), combined with two short cross-slope segments in correspondence of the shelf break. An overall distribution of θ, S, DO concentration, and neutral density indicates well defined layers of AASW, WW, and mCDW. AASW is situated in the relatively shallow surface layer (0 - 50 m depth), but with a wide range of temperature and salinity. Cold, fresh, and relatively uniform in temperature and salinity, WW lays beneath within the upper 400 m with larger thickness over the slope and shelf break. Beneath 400 m depth, warm and salty mCDW occupies the largest portion of the water column, decreasing its temperature and salinity approaching the shelf break (between positions C and D). There, it spreads over the continental shelf.

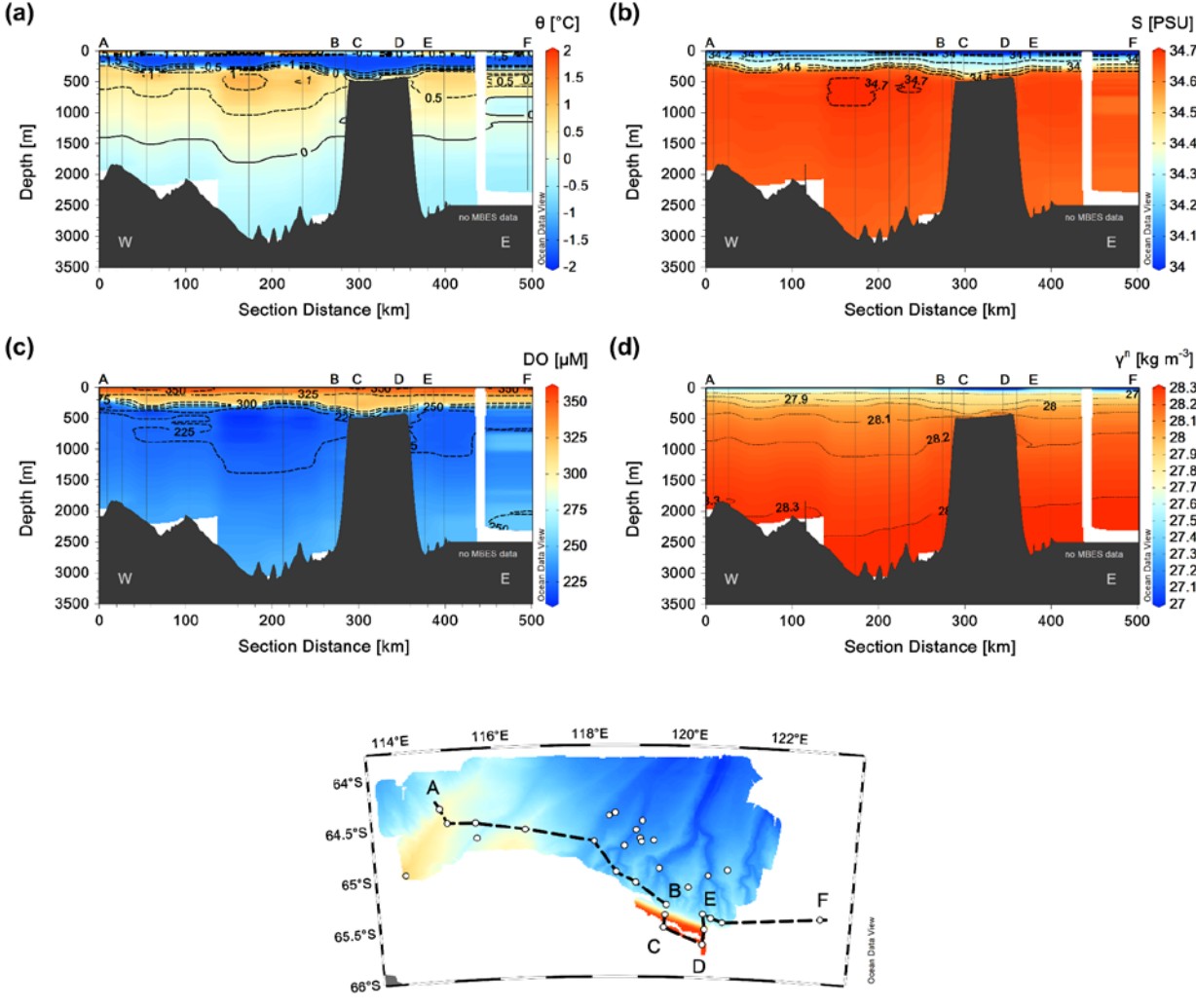

**Figure 4** - *Along-slope transect (AB, CD, EF) against distance, with two across-slope segments BC and DE. Vertical sections of θ (a), S (b), DO concentration (c), and neutral density (d) over the entire depth range. See the insert map for the location of the transect.*

A closer look into the upper 500 m layer reveals a well oxygenated AASW (Fig. 5). Its temperature and salinity decrease at the shelf break (area BCDE), probably due to the influence of cold and fresh ice melting water coming from the continental shelf (see Fig. 2b,c for the evolution of sea ice in January-February 2017). The portion of the water column occupied by WW, approximately between 50 m and 400 m depth, is cold ( -1.5°C < θ < -0.5 °C), relatively fresh (S < 34.45), and well oxygenated (DO >300 μM). The largest thickness of WW corresponds to the area approaching the continental shelf break (BCDE, water depths between 400

and 500 m), and it is typically associated with the downward tilting of density surfaces at the Fresh Shelf fronts, as described by
Thompson et al. (2018).

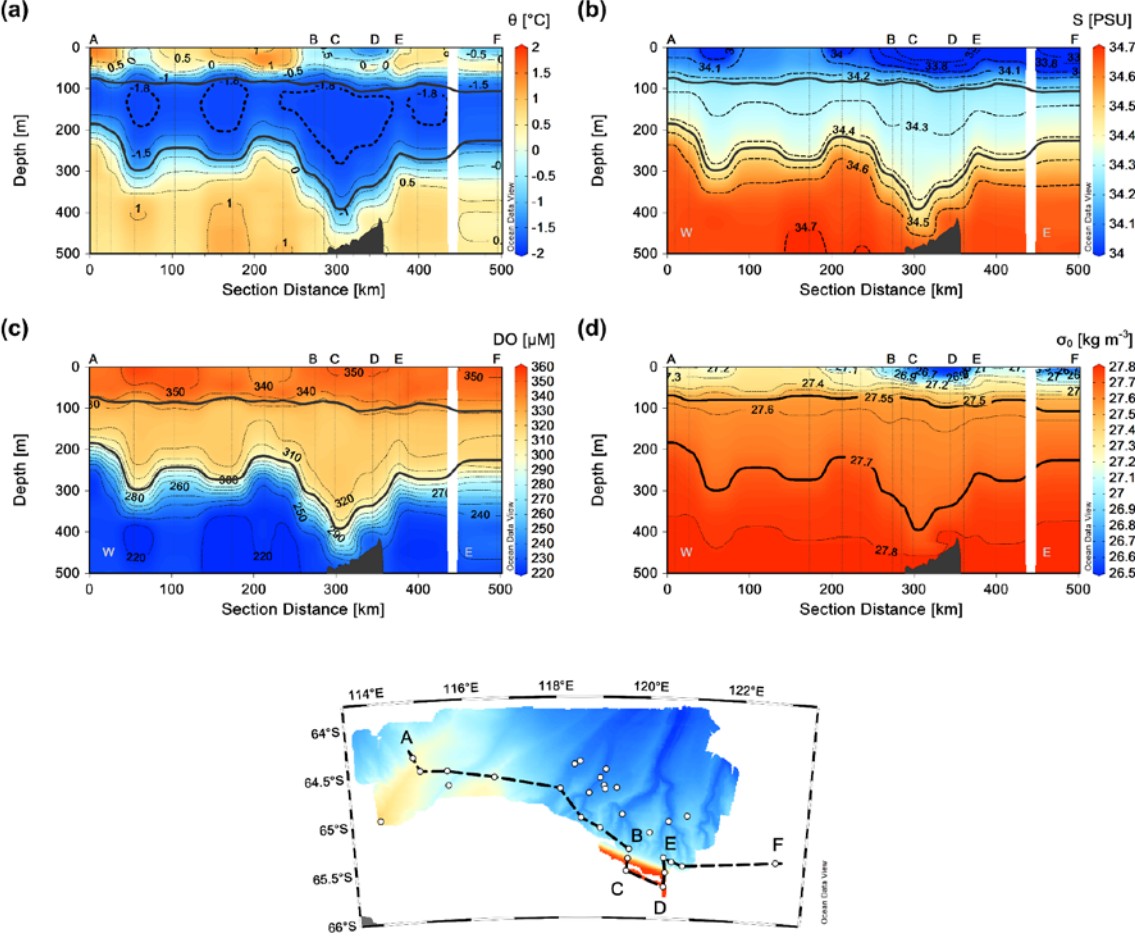

*Figure 5 - Along-slope transect (AB, CD, EF) against distance, with two across-slope segments BC and DE. Vertical sections of*
*θ (a), S (b), DO concentration (c), and σθ referring to 0 dbar (d) in the upper 500 m. Thick black lines in each panel indicate*
*isopycnals 27.55 and 27.70 kg m⁻³, delimiting the layer occupied by WW (defined according to Silvano et al., 2017). See the insert*
*map for the location of the transect.*
From 500 m down to the continental slope and rise area, to depths greater than 3000 m, both temperature and salinity progressively
decrease. Below 1500 m, in particular, cold waters have temperatures ranging from 0 °C to almost -0.5 °C near the bottom (Fig.
6). The lowest temperature values (-0.479 °C) are recorded in the easternmost part of the zonal transect, along with relatively low

salinity (38.638–38.640), and high oxygen (up to 248-250 µM) especially in the easternmost part of the section. The densest waters lay in the depressions/canyons/troughs between 118° and 119°E (neutral density 28.325 kg m⁻³). Despite the relatively large distance among CTD stations, the sections of the thermohaline properties and dissolved oxygen concentration reveal that, close to the bottom, in correspondence of the canyons and rugged bottom morphology (i.e., near positions B and E in Fig. 6) there are signals of possible pathways of dense waters. The high resolution of the neutral density isolines makes it possible to detect how the isopycnals, and all other properties, follow the bottom morphology (Fig. 6).

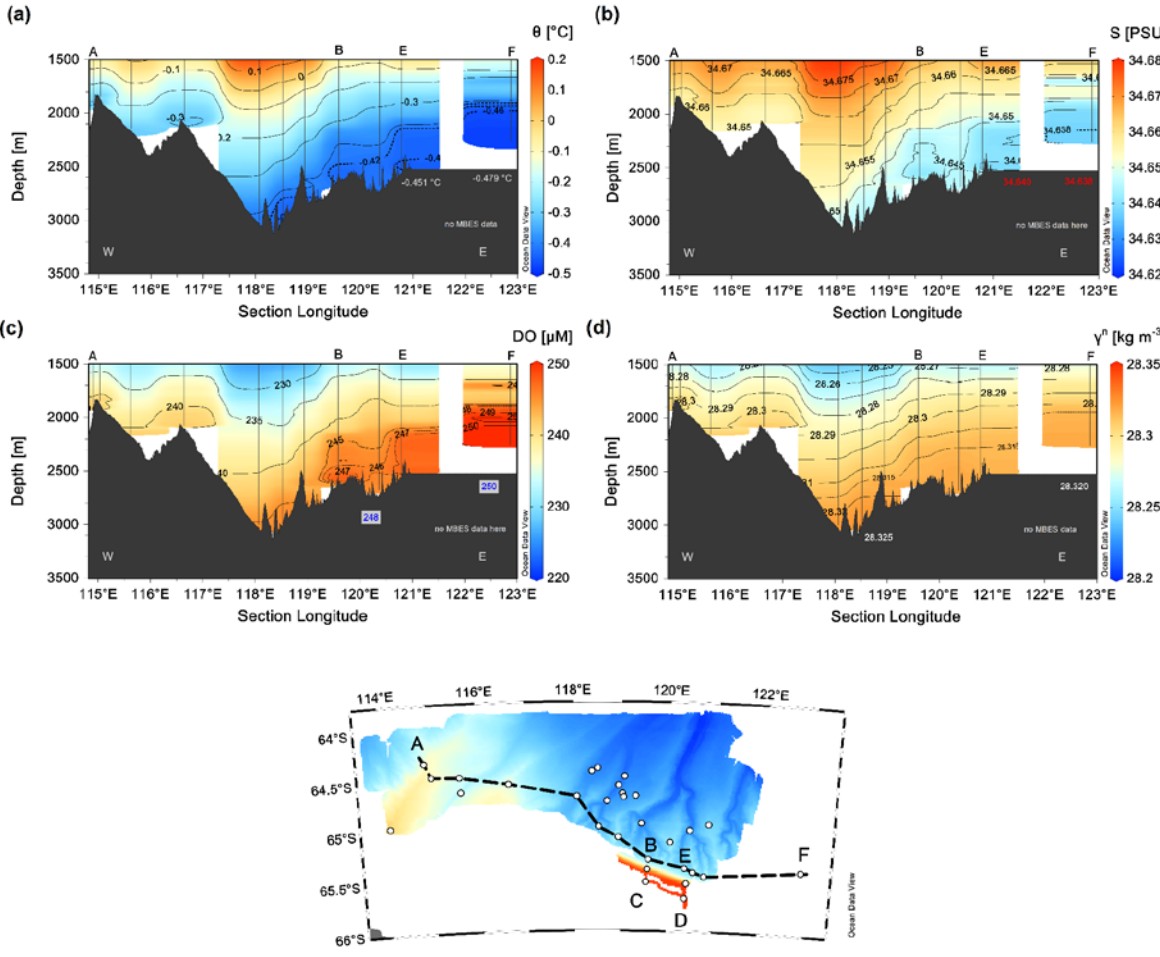

*Figure 6* - *Along-slope transect ABEF against the geographical longitude. Vertical sections of θ (a), S (b), DO concentration (c), and neutral density (d) in the deep depth range (1500 m – bottom). The isoline steps are chosen to make evident small changes of the hydrographic properties near the bottom where waters are the coldest, the freshest, the densest, as well as rich in dissolved oxygen content. See the insert map for the location of the transect.*

Patches of the deep and dense waters have properties that can be attributed to those of AABW (Figs. 3 and 4), which here seems
influenced by ALBW rather than RSBW. CTD profiles at the deepest stations, which are within the canyons, are slightly saltier
(see Fig. 3b) possibly due to the mixing either with the mCDW or with modified RSBW. From our data there is no clear evidence
about the contribution of dense waters originating from the continental shelf. However, these processes could occur only in
favourable conditions, e.g, during austral winters. The possible spread of such water out of the continental shelf area could give
origin to density-driven flows that descend within canyons along the continental slope and could help shape the seafloor
morphologies described by O'Brien et al (2020).

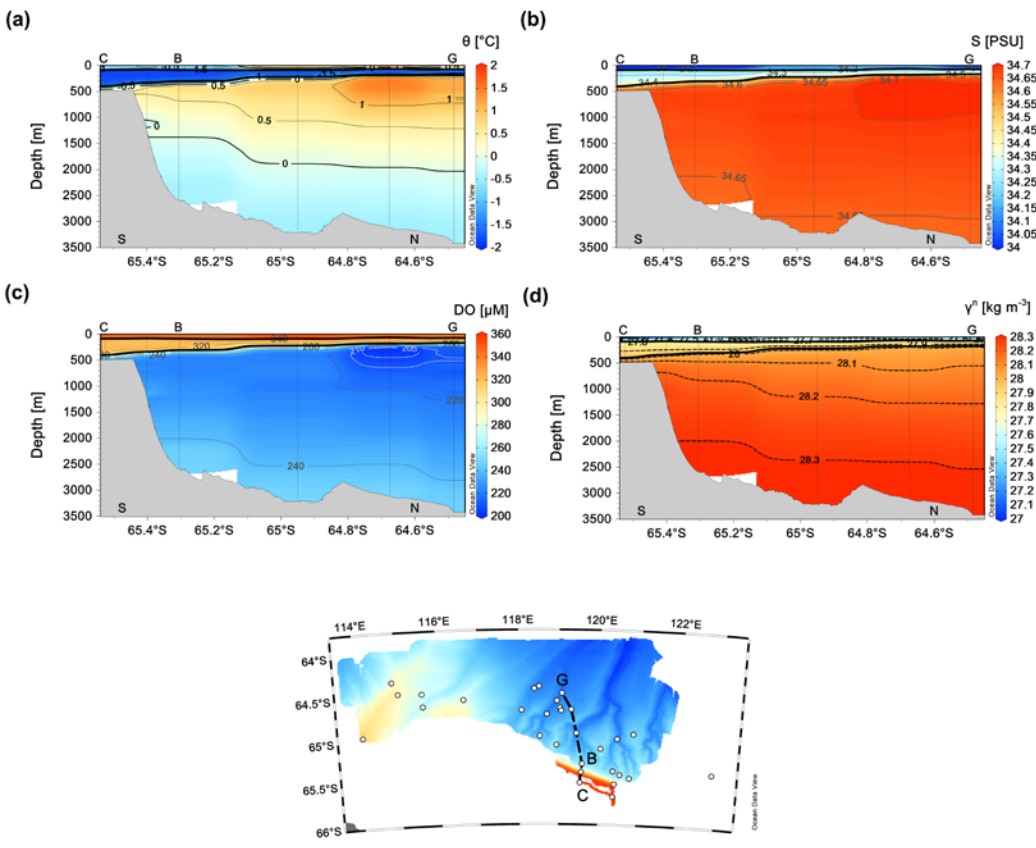


***Figure 7*** - *Across-slope transect CBG against the geographical latitude. Vertical sections of θ (a), S (b), DO concentration (c),*
*and neutral density (d) over the entire depth range. Isopycnals of 27.55 and 27.7 kg m$^{-3}$ are shown in all panels (thick black lines),*
*delimiting WW (according to Silvano et al., 2017). See the insert map for the location of the transect.*

Journal: https://www.earth-system-science-data.net/

To better identify the distribution of the water masses between the continental shelf and the off shelf area we draw a S-N across-
slope section. In fact, the transect CBG (see Fig. 7) illustrates a progressive thinning of the WW layer from the shelf break toward
the open sea, associated with the Antarctic Slope Front that separates cold and fresh shelf waters from the warm and salty mCDW
offshore (Thompson et al., 2018). Within the thick core layer of mCDW in the off shelf area, $\theta$ and S reach their maximum and
the DO content reaches its minimum between 64.6 °S and 64.8 °S. The layer occupied by mCDW progressively narrows in the
opposite direction toward the shelf break, where it also becomes less warm and less salty. There, the mCDW tongue protrudes
toward the shelf beneath the WW. On the continental slope, at approximately 1100 m depth, a slight temperature decrease appears
(Fig. 7a). However, the coarse space resolution between CTD stations precludes defining its origin. The deep and near-bottom
layers better illustrate how the isopycnals and associated thermohaline properties, along with the DO content, align with the
morphology of the sea floor (Fig. 8). There, coherent changes in $\theta$, S, and DO point out at a cold, relatively fresh, and oxygenated
bottom layer, approximately 400-500 m thick, leaning on the continental slope and rise. Its well delimited characteristics fade
away from 64.6 °S northward.

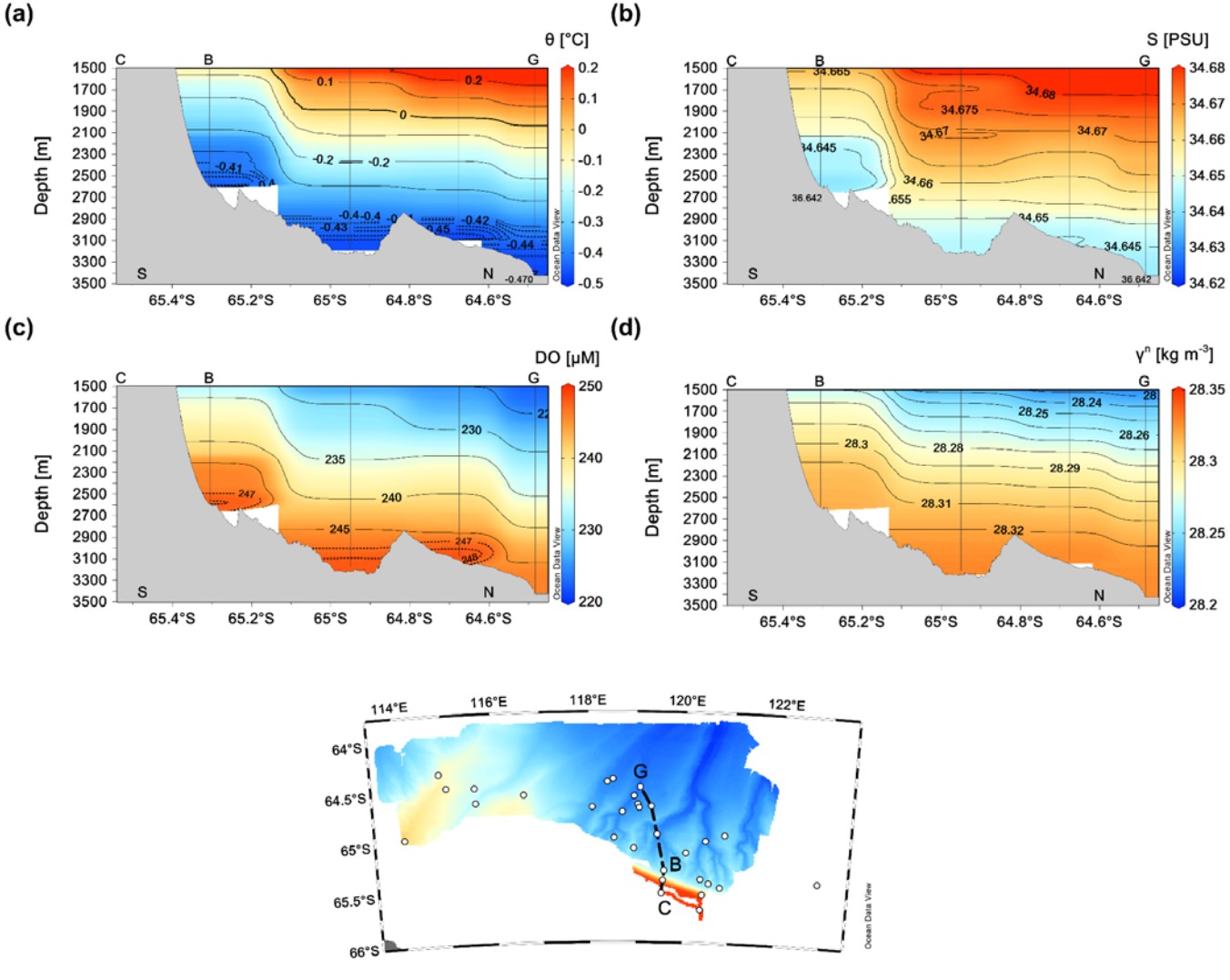


**Figure 8** - *Same as figure 7 but with a zoom into the 1500 m - bottom layer. The isoline steps are chosen to make evident small changes of the hydrographic properties near the bottom where waters are the coldest, the freshest, the densest, as well as rich in dissolved oxygen content. See the insert map for the location of the transect.*


The horizontal distribution of thermohaline properties near the surface (Fig. 9a, d, g, j) in the study area shows how they change in the across-slope direction, while they are more homogeneous along the direction parallel to the continental margin. From the shelf area, cold, fresh, and oxygenated waters approach the shelf break, where, due to their relatively low density, they mix with

other waters in the upper layer within the Antarctic Slope front. The strongest signal from shelf waters is visible at about 120°E
while fading in the northernmost portion offshore the study region, where higher values of θ and S (up to 2.0° C and 34.2,
respectively) are observed  at 15 m depth. The 450 m depth horizon (Fig. 9b,e,h,k) is chosen to represent the core of the mCDW
(θ values > 0) in the offshore area, within a layer that extends from about 300 m down to 2000 m (Fig. 7). Here, S reaches its
maximum values, and DO its minimum. In the region between 118°E and 120° E, the mCDW probably finds its favorable pathway
to the continental shelf due to the bottom morphology. Approaching the shelf break, the mCDW mixes with other water masses
originating locally or transported by the Antarctic Slope Current westward along the continental margin. Close to the bottom (Fig.
9c,f,i,l), maxima θ and S with values of about 0.5°C and 34.678, respectively, are registered over the shelf break at the southernmost
station 25 (at ca. 120° E, 420 m depth). These large near-bottom values reflect the mCDW impinging on the slope beneath the
cold and fresh WW. Overall, in the westernmost part of the study area (115°E - 117°E) at depths around 2000 m, the bottom layer
is occupied by waters with θ around -0.30°C and S around 34.65-34.66, hence slightly warmer and saltier (and less dense) than
bottom waters found at similar depths in the central and eastern sectors (Fig. 9c,f). Yet, the fact that bottom waters in the western
sector, which is generally shallower than the eastern one (Fig. 6), have lower DO values suggests that this area is also influenced
by mCDW. Intrusions of mCDW onto the continental shelf, forced primarily by wind-driven upwelling (Greene et al., 2017;
Silvano et al., 2019), can occur where the shelf break is deeper than 400 m. Hence, mCDW can flow down the landward sloping
continental shelf, reaching the Totten Glacier grounding zone (Greenbaum et al., 2015; Nitsche et al., 2017; Silvano et al., 2017;

293    2019).

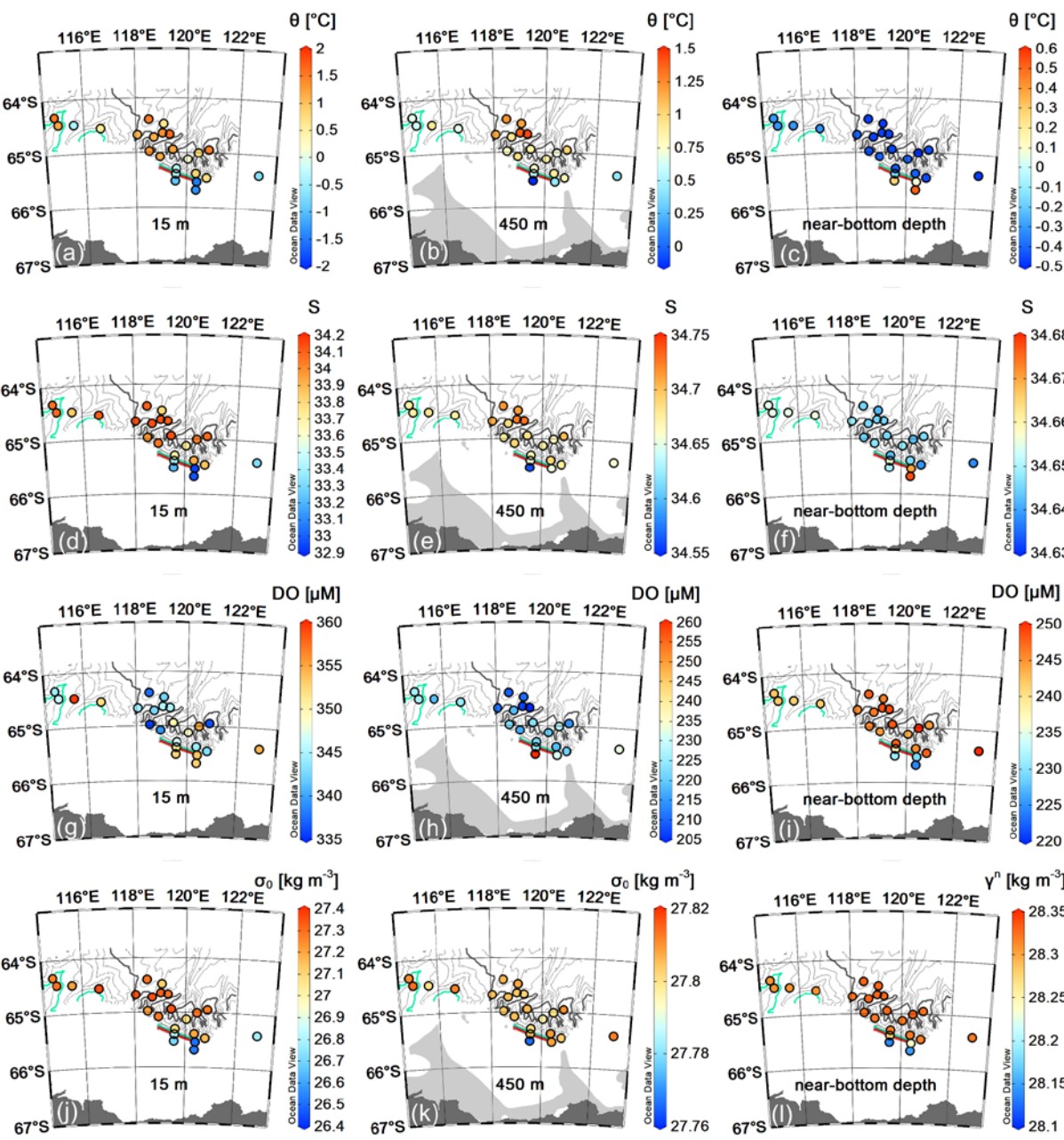

**Figure 9 -** *Near-surface (15 m depth), intermediate (450 m) and near-bottom θ (a, b, c), S (d, e, f), DO content (g, h, i), σθ (j, k) and neutral density γⁿ (l). The background bathymetry consists of isobaths (thin black curves) every 200 m obtained from MBES data acquired during the 2017 campaign; thick lines indicate isobaths at 1000 m (red), 2000 m (cyan), and 3000 m (gray).*

## 4 Summary and conclusions

Here we present oceanographic data (temperature, salinity, density, dissolved oxygen) collected offshore the continental margin of the Sabrina coast (East Antarctica) between 113°E and 122°E and 66° S and 64°S, from the shelf edge to the continental rise (Figs. 1 and 2). The water masses, described mainly according to their θ/S properties, are influenced by the seasonal heating and freshening of the uppermost layers, and their spatial distribution appears to be linked to the complex morpho-bathymetric setting of the study area. At about 400-500 m depth, the mCDW approaches shelf break in the area between 118 °E and 120 °E, around 64.5 °S, in good agreement with the numerical simulation results by Silvano et al. (2019). Small variations of the thermohaline properties in the 400–500 m thick bottom layer in correspondence of the canyons revealed, instead, that there are signals of dense waters with low θ and S values (-0.445 °C and 38.638 – 38.640, respectively) and relatively high DO values (240-250 µM). Densest waters, as shown in the θ-S plot (Fig. 3), have characteristics closer to typical ALBW than to RSBW. In general, mixing of these dense waters with the mCDW is not excluded. Data also reveal that bottom layers at stations with depths greater than 3000 m (i.e., inside canyons) are occupied by water slightly saltier than that found at depths between 2000 and 3000 m, but with similar temperatures. However, even though from our data there is no clear evidence about the contribution of dense waters originating from the continental shelf, we cannot exclude that during winter and spring seasons, occasional dense water plumes may descend and fill the deep layer, possibly mixing with the AABW. This hypothesis would support the fact that density-driven currents along the steep slopes in the eastern portion of this region could contribute, on the one hand, to shaping the present-day, deeply-incised canyons (Donda et al., 2020) and, on the other hand, to redistributing the different water masses (i.e., AASW, WW, and mCDW) off the continental shelf beyond the shelf edge, favouring their mixing. Our data also reveal that in the western sector of the study area (west of 118 °E) the bottom layer appears to be occupied by waters that are slightly warmer and saltier, as well as less oxygenated, than those found in the central and easternmost sectors. This difference in the deep water properties of the two sectors (eastern and western) seems to be an effect generated by the particular morphology of the seabed, being the western sector shallower than the eastern one, and hence more easily influenced by mixing with the mCDW. In addition, ocean currents and associated processes (along-slope flow, eddies, internal waves and tides, downslope density-driven currents) that involve local water masses may be different in the two sectors. This implies that mCDW intrusions (Greene et al., 2017) and thus heat transport on the Sabrina Coast continental shelf could be favoured by a combined effect induced by winds, eddies, and bathymetric constraints (i.e., along slope canyons, Hirano et al., 2021).

The IN2017_V01 expedition was conducted during austral summer, when rapidly changing sea-ice conditions also influence the near-surface and upper layers. The complex interplay of processes along the continental shelf, slope, and rise in this area is certainly challenging, suggesting the importance of learning more about oceanographic dynamics during both winter and summer seasons. In particular, the possibility of acquiring high temporal resolution data, especially during the winter period (e.g., by using long-term in situ observatories), could help in the understanding of the oceanographic processes that have contributed to shape the distal margin architecture and have influenced the evolution of the ice sheet.

**Data availability**

All data used in this work are archived in the enduring CSIRO Data Access Portal, https://data.csiro.au (direct link available at https://www.marine.csiro.au/data/trawler/dataset.cfm?survey=IN2017_V01&data_type=ctd, last access on October 26, 2021). Metadata records are made publicly available at http://www.marlin.csiro.au. Processed data and data products are, instead, publicly available through Data Trawler http://www.marine.csiro.au/data/trawler/index.cfm, the MNF web data access tool http://www.marine.csiro.au/data/underway/, and/or from national or world data centers most suitable for the dissemination of particular data types (https://doi.org/10.25919/yyex-t381). All metadata entries should list this requested acknowledgment statement where the data are presented or published (Armand et al., 2018).

**Declaration of Competing Interest**

The authors declare that they have no known competing financial interests or personal relationships that could have appeared to influence the work reported in this paper.

**Acknowledgements**

We thank the Marine National Facility (https://ror.org/01mae9353), the IN2017_V01 scientific party, led by the Chief Scientists L. Armand and P.E. O'Brien, MNF support staff and ASP crew members led by Capt. M. Watson for their help and support on board the RV Investigator. We also thank Andres Roubicek and Dave Watts (CSIRO NCMI, Information and Data Centre) for the assistance with data accessibility. This Project was supported through funding from the Australian Government's Australian Antarctic Science Grant Program (AAS #4333) and the Australian Research Council's Discovery Projects funding scheme (DP170100557). We would like also to acknowledge the financial support of the Italian National Antarctic Research Program (Programma Nazionale di Ricerche in Antartide, PNRA) that funded the participation of the Italian team to the cruise through the project TYTAN (Totten Glacier dYnamics and Southern Ocean circulation impact on deposiTional processes since the mid-lAte CeNozoic) led by F. Donda.

**Author contribution**

M.B., V.K., and F.D. conceived and wrote the article. M.B. and V.K. processed the data used to prepare the figures and performed the analyses. F.D. led the TYTAN project, L.K.A. and P.O. led the IN2017_V01 scientific cruise. L.A. contributed to collect experimental data and participated in writing the text. All authors contributed to the discussion and revision of the manuscript.

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
