# Peer review of "Water masses distribution offshore the Sabrina Coast (East Antarctica)"

_Earth System Science Data, 2021_

## Author Comment (AC1)

Bensi et al., present an interesting oceanographic dataset collected along the Sabrina Coast, nearby the Totten Glacier. This dataset can contribute to future oceanographic studies in this specific area as well as to basin scale studies. In my opinion this data description paper is well written and easy to read. Data acquisition methodology is well described, and data are easily accessible through the provided link. The authors also provide some analysis of the data, offering hypothesis and discussion points. As one of the strengths of this dataset is the scarcity of previous oceanographic data in the area, I suggest adding a new figure where the spatial distribution of the previously available data is compared to this new dataset.

*Answer: We thank the reviewer for her/his comments and suggestions provided, which we incorporated into the revised version, as detailed below. In particular, we have worked in the direction of providing a version of the paper that includes more references, improved figures, and some revised parts of the text. According to the Reviewer's suggestion, a new version of Fig. 1 now includes the spatial distribution of other available data.*

I have some minor comments that are reported below.

Line       15                    "The main water masses of the area"

   *Answer: Corrected*

Line       19                    Please resentence in order to simplify

   *Answer: the phrase was re-formulated. Now it is stated as follows:* "The latter (here we refer to AABW) is a mixture of dense waters from the Ross Sea and Adélie Land continental shelves. They are influenced by the mixing processes they undergo as they move westward along the Antarctic margin, also interacting with the Circumpolar Deep Water (warmer and saltier).

Line       33                    depends

   *Answer: since the verb refers to "Earth's climate processes as well as their future projections", we left the verb in the original form, i.e., "depend".*

Line       60                    Use the acronym for Sea-level equivalent

   *Answer: Corrected. SLE added in the text*

Line       119                   is the CTD equipped with a pump?

   *Answer: yes it is, it was a SBE9 plus V2 (pump-controlled) .*

Line       125                   Did you apply the standard CTD data treatment procedures?

   *Answer: Standard data processing and quality controls are reported in the mentioned reports, released by CSIRO Marine National Facility (https://www.cmar.csiro.au/data/reporting/get_file.cfm?eov_pub_id=1512). In*

*particular, CTD data processing was completed using CapPro processing software (Matlab software), version 2.9.*

Line    125             "Particular attention" I think this is a repetition of what previously stated

    *Answer: the reviewer is right, now it was corrected*

Line    131             Is the 0.002 psu an arbitrary target? Is it based on some bibliography?

    *Answer:* 0.002 psu is indicated as a standard accuracy within the requirements of Global Ocean Ship based Hydrographic Investigations Program (GO-SHIP). Technical notes from SEA-BIRD (https://www.seabird.com/asset-get.download.jsa?id=54663149001), instead, indicates initial accuracy for temperature (± 0.001 °C) and for Conductivity (± 0.0003 S/m).

Line    147             Please describe how MODIS images are used in the manuscript

    *Answer: more info has been added in this section. Now it is stated as follows:*

    *Finally, satellite images (MODIS - Moderate-resolution Imaging Spectroradiometer, Corrected Reflectance imagery) are used to highlight both the evolution of sea ice within the period covered by the IN2017-V01 cruise and the presence of the Dalton Polynya (Fig. 2), the open water surrounded by sea ice in the vicinity of Totten Glacier and MUIS. This is one of the largest Antarctic coastal polynya, with its wintertime average area of 3.7 ± 2.0 $10^3$ $km^2$ (6.5 $10^3$ $km^2$ at the time of the cruise; Fig. 2b,c), extending in the prevalent downwind direction (see e.g., Arroyo et al., 2019).*

Line    166             Please check symbols

    *Answer: checked, it was a problem occurred during pdf conversion.*

Line    182             Is the mixing the only possible process for this?

    *Answer: yes, we are confident that the water masses observed in our study area are a product of mixing that changes their original characteristics. However, there are several processes, such as combined effects due to along slope flows, eddies, internal waves and tides, downslope density-driven currents, which can involve local water masses and trigger their mixing. We now better stated this fact in the summary and conclusions.*

Figure  3               Please consider to use bigger dots for data

    *Answer: Figure 3 has been updated now, using bigger dots and showing endmembers for main water masses.*

[Figure]

| Line | 194 | "a wide range of temperature and salinity" |
|------|-----|---------------------------------------------|

*Answer: Corrected*

| Line | 195 | use acronym for Winter Water |
|------|-----|-------------------------------|

*Answer: Corrected*

| Figgs | 4,5,6,7,8 | Consider to add a map of the station to these figures |
|-------|-----------|--------------------------------------------------------|

*Answer: A map was added to these figures.*

| Fig | 4 | Why are you not using longitude for the x axis? |
|-----|---|--------------------------------------------------|

*Answer: In this representation, using longitude instead of distance on the x-axis would have caused the interpolated data to overlap, especially between points BC and DE, since the section does not zonally displace itself by crossing only longitude.*

Data Availability    The direct link to the data, as reported in the text would be useful here

*Answer: The direct link to the data was added (https://www.marine.csiro.au/data/trawler/dataset.cfm?survey=IN2017_V01&data_type =ctd)*

---

## Author Comment (AC2)

Bensi et al show new and exciting oceanographic observations on the continental slope region off the Totten Glacier. The Totten Glacier is experiencing high rates of basal melting and therefore understanding the processes that allow warm waters from the Southern Ocean to reach the continental shelf and ultimately the glacier is critical to better predict future sea level rise. The results shown by the authors are interesting as they provide the best observational view of the slope processes on the Sabrina Coast to date. They show how warm water approaches the coast and how canyons dynamics is important for cross-shelf exchange and bottom water formation/properties. I have one main comment, that in my view requires more analysis to be addressed, and some minor comments.

*Answer: We thank the reviewer for her/his comments and suggestions on the paper, which we incorporate into the revised version, as detailed below. In particular, we have integrated some important references that were missing and we implemented text and figures to make the article clearer. Finally, we carried out an additional analysis of the data as suggested by the reviewer in order to better distinguish the contributions related to water masses in the study area.*

**Major Comment**

- Line 219-250: I can see that changes in circulation are occurring within the canyons, but I am not sure if this is due to dense waters cascading down the shelf locally, or upstream, or if this is just a recirculation pathway of AABW due to the complex bathymetry. Also, Dense Shelf Water has never been observed on the Sabrina Coast continental shelf (neither in summer or winter).

What I suggest here is to "zoom" the TS plot to look at waters in the canyon. If you see some evidence of mixing with DSW (by simply inferring a mixing line between CDW and DSW), then this would suggest DSW formation.

*Answer: The text has been modified in the revised version of the paper, following the suggestion provided by the referee about the possible origin of the dense waters. Endmembers typical for the bottom waters in the region were added to the θ-S diagram in Fig. 3. This makes more clear the fact that bottom waters within our study region most probably originate mainly from ALBW that recirculates and mixes with local waters, following the complex bathymetric constraints. Based on our analyses, there is not clear evidence that at the time of the survey (Jan-Feb 2017) there was any contribution to the bottom waters coming from the Sabrina Coast continental shelf. We now better stated this concept in the paper, in particular in the introduction, which reports:*

"Ice loss can locally be also favoured by low levels of sea ice and Dense Shelf Water production in the Antarctic polynyas (Tamura et al., 2008), although there is still no clear evidence, surprisingly, of dense shelf water production in the Dalton polynya, east of the Totten ice shelf (Silvano et al., 2018)."

**Minor comments**

- Line 13-14: These data are new and incredibly important. However, other CTD data were collected on the Sabrina Coast both on the continental shelf and on the slope before 2017. I would re-phrase this sentence here and elsewhere.

*Answer: This sentence and the similar ones in the paper have been corrected. Moreover, we ameliorated Fig. 1, which now also shows areas where other already published hydrological data have been taken.*

[Figure]

- Line 28: "Downwelling" is probably not correct here. Maybe you are referring to gravity currents?

    *Answer: Corrected with "downslope gravity currents"*

- Line 48: a glacier to be susceptible to MISI needs not only to be marine based, but it also has to sit on a retrograde bedrock. The Totten Glacier at present is not clearly susceptible to MISI. I would just say that the Totten Glacier is susceptible to rapid ocean-driven melting.

    *Answer: The text was corrected as suggested. We have emphasized the key role of the ocean in driving glacier melting.*

- Line 50: Here I would refer to Rintoul et al (2016) as a reference for the water temperature reaching the Totten Ice Shelf.

    *Answer: Corrected*

- Line 57: Please check the basal melt rates of TIS and MUIS, as they seem too low here.

    *Answer: Checked, and corrected. Now it is stated as follows:*

    *However, the Totten Glacier is an exception, since the melting rates of this glacier and of the nearby Moscow University ice shelf (MUIS) are among the fastest in the East Antarctic Ice Sheets (Khazendar et al., 2013; Li et al., 2015; Mohajerani et al., 2018): the glacier draining into MUIS shows a 3 Gt/y loss in 1979–2003 and a 0.3 Gt/y gain in 2017, whereas Totten Glacier loss has increased through time from 5.7 Gt/y in 1979–2003 to 7.3 Gt/y in 2003–2017 (Rignot et al., 2019).*

- Line 61: add a reference for the temperature on the shelf. Also these estimates are based on snapshots, so the word "usually" is probably not well suited here.

    *Answer: Reference Added (Silvano et al., 2017), and the word "usually" was removed, since we agree that is a water mass definition based on data acquired during sporadic cruises.*

- Line 64: "in transporting ocean heat".

  *Answer: Corrected*

- Line 403: check the reference here.

  *Answer: The corrected reference here, is:* Silvano, A., Rintoul, S. R., Peña-Molino, B., Hobbs, W. R., Wijk, E. van, Aoki, S., Tamura, T., and Williams, G. D.: Freshening by glacial meltwater enhances melting of ice shelves and reduces formation of Antarctic Bottom Water, Sci. Adv. 2018;4 : eaap9467, https://doi.org/10.1126/sciadv.aap9467, 2018.

- Line 104-106: I agree that this is the best survey of Sabrina Coast slope region, but it is not the first one (e.g. Bindoff et al., 2000).

  *Answer: The reviewer is right. We have corrected this here and elsewhere in the text, added more references that were missing, and edited Fig1 to show the location of other surveys (with references) carried out in the study region.*

- Line 147-151: are you referring to the Dalton Polynya? If so, its size is much larger than what you report.

  *Answer: yes we refer to Dalton Polynya, and we added its name throughout the text. We have checked again its size by measuring it from satellite data (i.e., ca. 6.5 $10^3$ $km^3$), but also reporting data from:*

  *Arroyo, M. C., Shadwick, E. H., and Tilbrook, B.: Summer Carbonate Chemistry in the Dalton Polynya, East Antarctica, 124, 5634–5653, https://doi.org/10.1029/2018JC014882, 2019.*

- Line 171-172: in the study region I think AASW is fresh mostly because of sea ice melting not glacial (i.e. glacier) melting.

  *Answer: We agree, text was corrected*

- Line 175: please use neutral density for AABW as it is found below 2000 m depth.

  *Answer: Neutral density was added as parameter to define AABW*

- line 202-207: The deepening of the WW layer near the shelf break is due to the Antarctic Slope Front (e.g. Thompson et al., 2018; Rev Geophys.).

  *Answer: Corrected, and reference added*

- Line 235: "open sea area" -> "off shelf area" here and elsewhere.

  *Answer: Corrected*

- Line 241-245: also here you are describing the Antarctic Slope Front, a typical feature around Antarctica.

*Answer: Text was corrected*

- Line 243: "°N" -> "°S"

*Answer: Corrected*

- Line 258-262: please highlight whether you are describing surface or bottom properties. Note that mCDW is not usually found at 15 m depth. Near the surface sea ice melting often dominates the pattern of T and S during summer.

*Answer: this part was partially re-written to make the text clearer, since we refer to the near surface properties. We agree that mCDW does not reach 15 m depth. Now the text reads as follows:*

*"The strongest signal from shelf waters is visible at about 120°E in proximity of the shelf, while fading in the northernmost portion offshore the study region, where higher values of θ and S (up to 2.0° C and 34.2, respectively) are observed at 15 m depth."*

- Line 274: "where the seafloor is deeper".

*Answer: Corrected*

- Line 271-277: Note that the bottom depth changes between east and west (west is shallower). This can help explain the pattern of near bottom ocean properties you observe.

*Answer: Now this part has been partially re-written, including this concept here and in the "summary and conclusions".*

- Line 277: I would also cite Silvano et al (2019, jgr) where the authors describe MCDW intrusions onto the "Totten Trough".

*Answer: reference added*

---

## Author Response (AR2)

Thanks for the comments, we have now added (also in the paper) the DOI referred to the data presented in the manuscript (and acknowledge people from the data center for that). Moreover, we have updated authors' first name and last name as required. Finally, we have updated URSLs from www.cmar.csiro.au to www.marine.csiro.au, and added a ROR https://ror.org/01mae9353 for the Marine National Facility in the acknowledgements.

We hope that this version of the paper can be accepted for publication

Best Regards,

--
Dr. Manuel Bensi, PhD
Sezione di Oceanografia (oceanography section)
Istituto Nazionale di Oceanografia e di Geofisica Sperimentale - OGS
Borgo Grotta Gigante n. 42/c 34010 Sgonico (Trieste) - Italia